# A Response to Punti and Dingel's Critique of the Validity of the Intercultural Development Inventory for BIPOC Students. Comment on Punti, G.; Dingel, M. Rethinking Race, Ethnicity, and the Assessment of Intercultural Competence in Higher Education. *Educ. Sci.* 2021, *11*, 110

**Mitchell R. Hammer**

IDI, LLC, 2915 Olney Sandy Spring Rd, Olney, MD 20832, USA; mhammer@idiinventory.com

**Abstract:** Addressing social inequity and increasing intercultural competence is a critical challenge in the 21st century. This work should be informed by rigorous, scientifically grounded research, accurate interpretations of that research, and the implementation of policies and training that are based upon the integrity of such research efforts. The Intercultural Development Inventory (IDI), because of its psychometric integrity, is one such assessment tool that is used to pursue these challenges in higher education. The psychometric integrity of the IDI is unequivocally situated within the Standards for Educational and Psychological Testing (Standards for Educational and Psychological Testing 2014. American Educational Research Association [AERA], American Psychological Association [APA], and the National Council on Measurement in Education [NCME]). Punti and Dingel assert that the IDI is not valid specifically for Black, Indigenous, and People of Color (BIPOC) university students because it does not take into account the experience of being a minority/ethnic group member vis-à-vis racism and inequality. It is troubling that Punti and Dingel's critique (1) is based on their use of an interview methodology that does not comport with the Standards for Educational and Psychological Testing and (2) ignores the scientific evidence supporting the cross-cultural validity of the IDI with BIPOC.

**Keywords:** IDI; intercultural development inventory; intercultural competence assessment

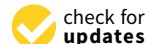



## 1. Introduction

Punti and Dingel (hereafter may be referred to as the authors) critique the validity of the Intercultural Development Inventory (IDI), specifically for Black, Indigenous, and People of Color (BIPOC), asserting that the IDI is not generalizable to BIPOC because it does not take into account the experience of being a minority/ethnic group member vis-à-vis racism and inequality. In my response, I will show that (1) Punti and Dingel's interview methodology is considerably flawed and their interpretations and conclusions are based upon conjecture and overgeneralization; and (2) the authors' critique ignores the scientific evidence supporting the cross-cultural validity and reliability of the IDI across 218,111 international and domestically diverse respondents, along with additional testing undertaken with 20,015 respondents who self-reported that they were members of an ethnic minority, which speaks to the validity of the IDI with respect to BIPOC students.

## 2. The Authors' Claim

Punti and Dingel claim, "the IDI has not been validated specifically for BIPOC in a way that considers statistical principles or triangulation of results with interviews. This weakness alone throws into question the IDI results, and what we can conclude from those results, for BIPOC" [1]. I present evidence that the validation of the IDI was undertaken consistent with psychometric standards of instrument development and that both individ-

ual interviews and focus group interviews were used throughout the validation process in ways that were inclusive of BIPOC perspectives and experience.

The authors also assert that, "a careful reading of the literature reveals that the IDI has been validated considering age, gender, education, and social desirability, but not race, social class, or ethnicity" [1]. I will review empirical findings showing that the validation of the IDI demonstrates its generalizability across social class/socioeconomic status differences, as well as racial/ethnic differences, vis-à-vis an individual's self-reported status as an ethnic minority.

Punti and Dingel further state that, "in sum, that the IDI has not been validated for BIPOC in the U.S. means that the experiences of BIPOC with racial inequality are not factored into the assessment, which potentially results in a white- or Eurocentric bias that downplays the role of racism in the daily lives of these individuals" [1]. I will clearly demonstrate the numerous psychometric protocols employed in validating the IDI that directly counter white or Eurocentric bias. Further, the authors identify as characteristic of being BIPOC the experience of racial inequality and racism associated with being an ethnic minority. I will review psychometric evidence indicating the applicability of the IDI to individuals who experience being an ethnic minority in a dominant culture (20,015 respondents) compared to dominant-culture respondents, which, again, the authors completely ignored in their critique.

### 3. Scientific Basis of Validation of the IDI

The development and validation of the IDI was based upon the set of standards for educational and psychological testing that was published in 2014 by the American Educational Research Association, the American Psychological Association, and the National Council on Measurement and Education (hereafter referred to as the standards) [2]. This set of standards is the gold standard for countering bias and supporting the development and validation of psychological testing. As stated, "the purpose of the standards is to provide criteria for the development and evaluation of tests and testing practices and to provide guidelines for assessing the validity of interpretations of test scores for the intended test use" [2].

The IDI has been subjected to extensive and rigorous validation protocols consistent with the standards for instrument development across globally and domestically diverse communities, including BIPOC [3–7]. The published articles and technical reports that delineate these protocols are readily available at www.idiinventory.com, accessed on 25 February 2022. Due to space limitations, I will highlight validation protocols and findings that are specifically relevant to the validity of the IDI to BIPOC's experience as an ethnic minority and also share results from testing across socioeconomic status/social class differences.

The authors incorrectly claim that, "the IDI has not been validated specifically for BIPOC in a way that considers statistical principles or triangulation of results with interviews" [1]. This statement is problematic for a number of reasons.

First, 20,015 BIPOC respondents who reported their identity as an ethnic minority were represented in the larger (218,111 respondents) validation sample. Targeted validation testing was conducted for generalizability of the IDI to these self-reported ethnic minority respondents. Additional testing across other subgroups by gender, age, and education level/social status was also completed. It should also be noted that 150,577 respondents were within the educational sector, and the remaining 67,534 respondents were within the organizational sector, with validation testing again including both subgroups, as well as validation testing within each of these subgroups separately.

Second, as will be demonstrated below, qualitative individual interviews and focus group interviews were extensively used to validate the IDI's applicability and inclusion of the diverse experiences and perspectives of BIPOC, and these interview protocols were consistent with the standards.

Third, Punti and Dingel's own interview method and the manner in which interpretations and conclusions were drawn are not supported in the standards.

Fourth, the authors mistakenly suggest that their interview approach represents a valid method for assessing the similarity or lack of similarity between the discourse gathered from their brief interviews and the respondents' identified developmental orientation from the IDI. Later in this response, I will return to this issue regarding the fallacy of asserting the primacy of their interview analysis over the scientifically derived orientation identification from the IDI.

## 4. Inaccuracies in the Authors' Interview Methodology Approach to Critiquing the Validity of the IDI with BIPOC

Punti and Dingel made no attempt to review the psychometric evidence that supports the validity and reliability of the IDI in general or specifically with BIPOC in spite of the fact that IDI, LLC makes this evidence transparent and easily accessible on their website (idiinventory.com, accessed on 25 February 2022).

- The authors made no documented attempt to evaluate their interview approach with interview approaches recommended by the standards.
- Doing so would have revealed that (1) strong evidence exists confirming the validity of the IDI to BIPOC and that (2) Punti and Dingel's interview strategy is not a recognized methodology for testing the validity of psychological assessments, including the IDI.

A deeper look at Punti and Dingel's interview methodology reveals the following:

1. The authors report that they themselves conducted 34 semi-structured interviews with first-year college students who had just completed their first semester on campus and who completed the IDI on arrival approximately two months earlier. Each interview lasted between 30 min to 60 min. The interviews were recorded and transcribed.

However, these 34 interviews were not the sample for their critique article [1]. That is, the actual student sample used in their article included only students who profiled on the IDI in denial or polarization. Given their focus on BIPOC students, only two BIPOC students' responses were actually analyzed who had the denial orientation, and only two BIPOC students were analyzed in the polarization orientation. Unanalyzed for reasons unknown were 10 BIPOC student responses in minimization and 4 BIPOC student responses in acceptance. Their critique, therefore, that the IDI is not supported with BIPOC is based on interviews of only 4 BIPOC-identified students!

2. While each interviewer analyzed each of the 34 interview transcripts, each interview was independently coded by the two authors based on responses to the interview guide. A core untested assumption of Punti and Dingel is that by simply asking students a series of limited, open-ended questions, they, as both interviewers and interview transcript coders, are able to accurately diagnose the students' underlying orientation towards differences.

In order to confirm the accuracy of such an assumption, Punti and Dingel would need, at a minimum, to demonstrate acceptable levels of inter-rater reliability and articulate various protocols they used to control for inaccurate diagnosis of the student's developmental orientation from their interview sample. No such inter-rater reliability was undertaken by Punti and Dingel, nor were any controls for any threats to the validity and accuracy of their interpretations and conclusions identified or implemented.

3. One important factor particularly relevant to interpretations and conclusions drawn from interview data gathered to assess/diagnose an individual's level of competence is results from research conducted across multiple fields that have consistently revealed that people often overestimate their level of capability. So named the Dunning–Kruger effect, this is a cognitive bias in which people generally believe they are more competent than they really are [8].

This research is important to take into consideration when interviewing people about how they engage differences because it points to the fact that asking people about their

experience around difference is neither a reliable nor accurate strategy for determining how interculturally competent they actually are. That is, asking people questions about their views, perspectives or experience around difference can reflect the Dunning–Kruger effect rather than their actual level of intercultural competence (i.e., developmental orientation).

Although Punti and Dingel interpreted interviewees' discourse as indicative of orientations further along the intercultural development continuum than the orientation empirically identified by the IDI (in this case, a denial or polarization mindset), it is more likely the authors' interpretation is reflective of the Dunning–Kruger effect. Punti and Dingel did not include any protocols to address such concerns.

In contrast, the validation protocols of the IDI do take this information into direct consideration. That is, the IDI measures both the respondent's perceived orientation (the orientation from which they believe they engage differences) and the respondent's developmental orientation (the orientation from which they actually engaged differences). The presence of the Dunning–Kruger effect is empirically assessed by the IDI as the gap between perceived orientation and the development orientation.

4. There are other compelling alternative explanations for the "meaning" of the discourse gathered from the interviews than what Punti and Dingel present. For example, the interview discourse could be reflective of the influence of the power dynamic between the professors' status as interviewers and the interviewees' status as students and completely unrelated to the students' orientations toward differences. Were the students simply reflecting back to their professors what they thought the professors wanted to hear? The authors did not describe any protocols to address such concerns. In contrast, the IDI empirically controls for concerns such as this by testing for social desirability, which was found to have no significant impact on respondent scores.

5. The interview discourse could also be reflective of the fact that the students were interviewed two months after they completed the IDI and during their first semester on a college campus where they were living away from their home, likely for the first time. The first semester of the college experience is often a "difference-intense" experience for first-year college students. We know, for example, from study-abroad research [9], that such "difference-intense" experiences can significantly change students' developmental orientation (as measured by the IDI). The authors included no protocols to account for any intercultural development that may have taken place from the moment they arrived on campus to the time, two months later, that they were actually interviewed. The IDI presents a valid and accurate measurement of intercultural competence along the intercultural development continuum (IDC) at the time that the IDI is completed. Given the "difference-intense" experience of moving to an unfamiliar college environment, at a minimum, the authors could have readministered the IDI to their interview sample at the time they actually conducted the interviews.

6. Punti and Dingel report that when discrepancies arose in the coding of the interview transcripts, they were discussed until consensus was reached. What the discrepancies were, how often discrepancies arose and how consensus was reached is unstated. There were no inter-rater reliability statistics computed, which would have provided a baseline measure of the degree of interpretive consistency exhibited by the authors. Again, the authors did not include any protocols in their article to address these concerns.

The point is that one does not know what the meaning of the discourse is when gathered through the authors' interview protocols. To suggest that such unsupported diagnosis of the interviewees' developmental orientation by Punti and Dingel is valid and takes precedence over the psychometrically derived diagnosis from the IDI is specious.

## 5. Use of Interviews in Validation of the IDI

Let me address, at this point, the specific IDI validation efforts that were employed vis-à-vis the use of interviews and BIPOC (It should be noted the "standards" include

appropriate interview strategies and interpretation guidelines which were followed in the validation of the IDI). Interviews were used that spoke to and captured the unique experiences around international and domestic differences, including those of BIPOC [3]. These include:

- Interviews were used in the initial analysis of how diverse individuals experience cultural difference and the relationship of those experiences to the underlying theoretical framework of the intercultural development continuum. Domestic (US; including BIPOC) and international perspectives and experiences among interviewees and interviewers were represented in these initial interviews. Results from these interviews found that the intercultural development continuum provides a rich and accurate explanatory framework for understanding how cultural difference is experienced by individuals from diverse backgrounds and communities.
- Interviews were used to generate the actual statements of diverse individuals regarding how they experienced differences in terms of the continuum. These interviews were transcribed and analyzed by culturally diverse teams, which again included BIPOC. Inter-rater reliability was calculated in order to ensure that the statements being captured were consistently identified across diverse interviewers/raters.
- This use of interview-based transcripts resulted in a pool of potential items coded by a diverse group of raters in which the pool of items generated did not reflect the perspective or possible bias (or power differences) of myself, the lead researcher. In other words, the items were grounded in the diverse experiences of international and domestic interviewees, not, as the authors mistakenly assert, [from] "a white- or Eurocentric bias that downplays the role of racism in the daily lives of these individuals" [1].
- Multiple group interviews, again with interviewees drawn from both international and domestically diverse communities, were conducted to ensure that the meaning of each of the statements was the same across diverse perspectives and diverse experiences. Items that interviewees interpreted differently were eliminated from the pool of potential items as a result of these focus group interviews.
- The expert panel review method was used, and inter-rater reliability analysis was conducted to ensure these items are generalizable and similarly understood across a wide band of international and domestically diverse groups.

Overall, extensive use of interviews consistent with the standards were used to validate the IDI in ways that were inclusive of diverse perspectives and experiences, including those of BIPOC.

## 6. The IDI Has Been Validated with Respect to Race, Social Class, and Ethnicity

The authors claim the IDI is not validated regarding race, social class, or ethnicity. "A careful reading of the literature reveals that the IDI has been validated considering age, gender, education, and social desirability, but not race, social class, or ethnicity" [1]. The authors go on to assert that, "in sum, that the IDI has not been validated for BIPOC in the U.S. means that the experiences of BIPOC with racial inequality are not factored into the assessment, which potentially results in a white- or Eurocentric bias that downplays the role of racism in the daily lives of these individuals" [1].

In fact, multiple analyses, consistent with the standards, were undertaken in 2017 by ACS Ventures, a highly reputable instrument-assessment organization, to complete a series of additional independent psychometric analyses to further investigate the performance of items and scores from the IDI (i.e., to further test the cross-cultural validity of the IDI, including specific analyses around social class and respondents' experience as an ethnic minority [7].

Their exhaustive testing of the IDI was undertaken with 218,111 IDI respondents from a wide range of national and international cultural communities. This included samples of 67,534 organizational sector respondents and 150,577 educational sector respondents.

Key variable comparisons were conducted examining whether the IDI items and scales systematically varied by each of the dimensions listed below in ways that would limit the applicability of the IDI and whether there was cultural bias. The key variables examined in this research that are directly applicable to the claim that the IDI is not sensitive to BIPOC minorities' marginalized experience or to social class difference are: (1) identifying significant differences in respondents' IDI item and scale scores (i.e., Perceived Orientation and Developmental Orientation scores) between respondents who identified themselves as an ethnic minority in their country (20,015 respondents) or members of the majority (81,814 respondents) and (2) identifying differences in respondents' IDI item and scale scores by educational level. In this case, educational level was used as a surrogate indicator of social class differences. This is consistent with research findings that have indicated very high correlations between educational level and socioeconomic status. The American Psychological Association (APA) identifies education level as a reliable indicator of socioeconomic status [10].

Results from ACS Ventures' independent research indicate that differences in total scores between respondents who have ethnic minority or majority status are small and not consequential. Differences between more educated and less educated respondents (social class differences) reveal similar results. These results empirically support the conclusion that the IDI is a cross-culturally valid assessment of intercultural competence both internationally and domestically and generalizable to BIPOC. Similar support was obtained across gender, age, and country differences as well.

The next set of analyses conducted by ACS Ventures focused on differential item functioning (DIF). In a DIF analysis, the performance of respondents on each item is reviewed to evaluate whether the item appears to unfairly favor one group over another. DIF analyses were completed comparing the performance across male and female students between respondents who self-identified as an ethnic minority/majority in their country and based upon education level. Overall, these findings did not identify any items with notable DIF by gender, ethnic majority/minority status, or education level/social class. This strongly supports the conclusion that the IDI does not contain cultural bias and is generalizable across gender; a wide range of international and domestic differences, including BIPOC; and socioeconomic status.

## 7. Conclusions

Helping individuals in our educational institutions and our organizations more effectively address social inequity and increase intercultural competence is one of the most important challenges of the 21st century. It is critical that this work be informed by rigorous, scientifically grounded research, accurate interpretations of that research, and the development of policies and training that are based upon the integrity of such research efforts.

The Intercultural Development Inventory is one such assessment tool that is being used to pursue equity, address racism, and build intercultural competence. One reason people have embraced the use of the IDI is because of its psychometric integrity and their own success in designing IDI-guided development strategies to achieve goals around diversity, equity, and inclusion.

The validity and reliability of the Intercultural Development Inventory is fundamentally situated within the Standards for Educational and Psychological Testing. All such instruments that are developed based upon the standards recognize that validation efforts of any instrument are ongoing but guided by rigorous standards [2]. Such has been the case with the Intercultural Development Inventory. However Punti and Dingel's critique [1], which asserts that the IDI has not been validated with BIPOC, is unsupported, misinformed, and unacceptable. It is hoped that the reader of this response will find the empirical information presented useful to their own work with the IDI in addressing social inequality and building intercultural competence.

**Funding:** This research received no external funding.

**Institutional Review Board Statement:** Not Applicable.

**Informed Consent Statement:** Not Applicable.

**Data Availability Statement:** Data available in a publicly accessible repository that does not issue DOIs. Publicly available datasets were analyzed in this study. This data can be found here: https://idiinventory.com/idi-validation/, accessed on 25 February 2022.

**Conflicts of Interest:** The author declares no conflict of interest.

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
