# Peer review of "A Response to Punti and Dingel’s Critique of the Validity of the Intercultural Development Inventory for BIPOC Students. Comment on Punti, G.; Dingel, M. Rethinking Race, Ethnicity, and the Assessment of Intercultural Competence in Higher Education. Educ. Sci. 2021, 11, 110"

_education, doi:10.3390/educsci12030176_

Round 1
Reviewer 1 Report
This is a quality response to the questioning of the validity of the IDI. This discussion reminds me of my first graduate level statistics professor who came to academia after working for years as a psychometrician for College Board. He once shared that it was always a challenge to provide a reliable and valid standardized normed assessment that will always satisfy everyone. Even when you follow all the rules and critieria when constructing assessments, there will be groups who argue that their group was not validated when constructing the instrument. The authors here followed the rules and criteria. Having shared this, I do think when even poorly done qualitiative research conflicts with well done quanitative research, it provides the opportunity to conduct a future study that experimentally tests the concern raised by the quantitative study. In this case, we can add a brick to our wall of knowledge by conducting solid future research using the IDI with an adequate sample size of the group under consideration.
Author Response
Thank you for this insightful review of the submitted article, as well as your suggestion for possible future studies. Based upon your review, we have not included any substantive changes to the original submission, although we have done an additional review for minor grammar edits. We appreciate your comments and look forward to publication. However, we remain available to make any requested edits in the future.